# BlockDFL: A Blockchain-based Fully Decentralized Peer-to-Peer Federated Learning Framework

## ABSTRACT

Federated learning (FL) enables collaborative training of machine learning models without sharing training data. Traditional FL heavily relies on a trusted centralized server. Although decentralized FL eliminates the central dependence, it may worsen the other inherit problems faced by FL such as poisoning attacks and data representation leakage due to insufficient restrictions on the behavior of participants, and heavy communication cost, especially in fully decentralized scenarios, i.e., peer-to-peer (P2P) settings. In this paper, we propose a blockchain-based fully decentralized P2P framework for FL, called BlockDFL. It takes blockchain as the foundation, leveraging the proposed PBFT-based voting mechanism and two-layer scoring mechanism to coordinate FL among peer participants without mutual trust, while effectively defending against poisoning attacks. Gradient compression is introduced to lowering communication cost and prevent data from being reconstructed from transmitted model updates. Extensive experiments conducted on two real-world datasets exhibit that BlockDFL obtains competitive accuracy compared to centralized FL and can defend poisoning attacks while achieving efficiency and scalability. Especially when the proportion of malicious participants is as high as 40%, BlockDFL can still preserve the accuracy of FL, outperforming existing fully decentralized P2P FL frameworks based on blockchain.

## CCS CONCEPTS

• **Computing methodologies** → **Distributed artificial intelligence**; **Learning paradigms**.

## KEYWORDS

Decentralized Federated Learning, Peer-to-Peer, Blockchain, Trustworthy Federated Learning.

**ACM Reference Format:**
Anonymous Author(s). 2018. BlockDFL: A Blockchain-based Fully Decentralized Peer-to-Peer Federated Learning Framework. In *Proceedings of Make sure to enter the correct conference title from your rights confirmation emai (Conference acronym 'XX)*. ACM, New York, NY, USA, 11 pages. https://doi.org/XXXXXXX.XXXXXXX

## 1 INTRODUCTION

Federated learning (FL) [24] enables multiple mobile and Web-of-Things devices to jointly train a machine learning model while keeping the data on their own devices [31]. It can contribute to a series of web applications such as web tracking based on FL [36]. However, traditional FL heavily relies on a trusted centralized server, where the bandwidth limitations caused by long-distance transmission restricts the application of FL in large-scale and highly complex problems. The central dependence can be solved by decentralized FL [6, 14, 15, 44]. However, the primary challenge faced by decentralized FL is the trust between participants, especially in collaboration across business organizations. In a scenario without mutual trust, a global model may not be trustworthy, and the contribution of different participants is hard to authoritatively quantified.

Blockchain, a distributed ledger originated from decentralized currency systems, offers distributed trust that enables cooperation among participants without mutual trust by forcing the participants to behave honestly [17, 34, 44]. It can also record stake for monetary reward to motivate honest behaviors [47], since decentralized systems relieve the burden of maintaining centralized servers for the operators. These characteristics make blockchain an promising basis for designing a decentralized trustworthy FL framework.

Unfortunately, decentralization also exacerbates the inherent issues of FL such as 1) vulnerability to poisoning attacks; 2) relatively insufficient privacy protection because the private training data can be reconstructed from intermediate model updates by *model inversion attack* [12, 28, 48]; and 3) inefficiency caused by heavy communication cost for transmitting model updates. Existing blockchain-based FL frameworks usually integrate additional protection mechanisms to partially solve the privacy and security issues [1]. For example, protecting the privacy by differential privacy (DP) [19, 26, 45, 46], homomorphic encryption [2, 34] and secure aggregation [23, 26], and ensuring the security by Krum [3], threshold-based testing [39] and auditing [2]. However, existing frameworks still have some limitations on technical selections. For privacy, although DP provides provable protection [33], it lowers model accuracy. Homomorphic encryption and secure aggregation bring tremendous computation and communication cost, lowering the efficiency. For security, existing approaches mainly apply Krum on local updates [26, 32, 45], ignoring that global updates may also be poisoned. Besides, Krum performs not well when facing not identically and independently distributed (non-IID) data. Threshold-based testing relies on manual thresholds that are not easily to be determined. Auditing provides only traceability but no defenses. Moreover, existing solutions often neglect efficiency optimization. Some of them rely on mining for consensus that further deteriorates the efficiency [5, 17, 32] due to a large number of hash calculations. Additionally, some of existing frameworks are not fully decentralized [6, 14, 16, 20, 29, 37, 45], i.e., relying on a global trust authority or trusted servers [26]. Thus, there needs *a fully decentralized peer-to-peer (P2P) solution with high efficiency that protects FL systems from being poisoned and the training data from being reconstructed.*

To address these issues, we propose BlockDFL, a **fully decentralized P2P FL framework** based on blockchain that protects the privacy in terms of data representation and security in terms of poisoning attacks while achieving high efficiency. To effectively filter out poisoned updates even in non-IID setting, we propose a two-layer scoring mechanism, where local updates are filtered according to the stake and scored by median-based testing, and global updates are scored by Krum. To uniquely select a global update in each round without forking, we design an efficient voting mechanism based on the Practical Byzantine Fault Tolerance (PBFT) [4] algorithm. The two mechanisms work jointly to defend poisoning attacks. Gradient compression is introduced to protect the training data from being reconstructed and further reduce communication cost. The main contributions of this paper are threefold:

- We propose BlockDFL, an efficient decentralized P2P FL framework with blockchain as the foundation, which provides security and privacy protections for FL while achieving high efficiency.
- To reach the consensus on one suitable global update in each round of FL, we propose a PBFT-based voting mechanism, which never forks. It works together with the proposed two-layer scoring mechanism to fulfill poisoning-resistance.
- We implement a prototype of BlockDFL and conduct extensive experiments on two real-world datasets, showing that BlockDFL achieves good efficiency and scalability and can resist poisoning attacks when there are up to 40% malicious participants for both IID and non-IID data. We also experimentally demonstrate that there exists appropriate sparsity that protects data representation privacy without harming the accuracy and security of BlockDFL.

## 2 PRELIMINARIES

### 2.1 Poisoning Attack

FL suffers the risk of poisoning attacks. Malicious participants can upload poisoned models to negatively impact the convergence of FL [22], such as wrongly labeling the training data within one certain class and training the local model on the tampered datasets, causing the global model unable to distinguish the data of this class.

There are many defenses suitable for a centralized system [3, 22, 38]. For example, Krum [3] regards the model updates significantly differ from others as poisoned ones. However, in a P2P system where there is no mutual trust among participants, it is hard to decide which participant should be responsible for detecting poisoned model updates, and participant may distrust the judgments from the others. BlockDFL solves these problems by a two-layer scoring mechanism, i.e., in each round, the local updates are scored through local inference by several other participants (aggregators in Section 3) to form up several global updates, then these global updates are scored through Krum by another group of participants (verifiers in Section 3) to finally select one global update.

### 2.2 Data Representation Leakage

Existing studies show that the model updates shared by participants in FL still contain some information of training data useful for data reconstruction through model inversion attack [12, 28, 43, 48]. If the model updates are leaked, attackers can reconstruct the private training datasets [26, 34]. Thus, FL needs further privacy protection since there is no protection for model updates in vanilla FL [24],

especially in decentralized systems since the model updates are transmitted among ordinary participants which may be malicious.

Such kind of risk can be defended by gradient compression that only transmits elements with large absolute values in model updates [48]. Attackers cannot reconstruct data from sufficiently sparse updates [27, 28]. Intuitively, excessive compression may bring negative impact on Krum since it filters out model updates heavily differ from the direction of the majority of updates. But we experimentally find that many of the indexes of the transmitted elements with the largest absolute value in different updates may still overlap, enabling to spatially distinguish the normal and malicious model updates with sparsification introduced. Besides, it may even improve the accuracy of Krum since it helps to focus only on important parameters, which may reduce the negative impact of unimportant parameters on the element-wise distance calculation. More details about this are available in Appendix A.1.

BlockDFL drops over 90% and 85% elements with lower absolute value in model updates before transmission respectively on two datasets so that model inversion attacks represented by the Deep Leakage from Gradients (DLG) attack [48] cannot obtain any useful information as experimentally demonstrated in Appendix A.2.

## 3 BLOCKDFL OVERVIEW

BlockDFL is designed for decentralized P2P FL with the following goals: 1) to prevent the global model from being jeopardized by poisoning attacks, 2) to prevent the private training data from being revealed, and 3) to conduct FL efficiently. As shown in Fig. 1, there are four processes in it during each communication round: 1) *Role Selection*, 2) *Local Training*, 3) *Aggregation* and 4) *Verification and Consensus*, some of which contain more than one steps.

It is assumed that participants can obtain the public key for verifying digital signatures of the others and send information through broadcasting. The stake recorded on the blockchain can be tied to monetary reward from mobile operators or AI service providers, since a decentralized FL system relieves them of the heavy burden of setting up and maintaining a centralized server. Thus, as in [13, 26], it is reasonable to assume that participants holding large amounts of the stake tend to perform obligations honestly, because they can benefit more from the monetary reward.

Participants are granted with three different roles, i.e., *Update Provider*, *Aggregator* and *Verifier*. The update provider is for training a model based on its private training data and sharing its local update to aggregators. It works independently. The aggregator is responsible for collecting local updates and selecting a certain number of them for aggregating global update. It works independently, too. The verifiers preside over electing a suitable global update together and packaging it with the digital signatures created by the verifiers' private keys and the identity of its aggregator and update providers into a block newly added to the blockchain. They score global update independently, and select one global update collaboratively. The independent and cooperative steps are marked with different colors in Fig. 1. In BlockDFL, adding a block means that all participants have conducted a round of communications (equivalent to executing the FedAVG algorithm [24] once in FL). If the block is not empty, all participants will update their model according to the global update contained in the newly added block.

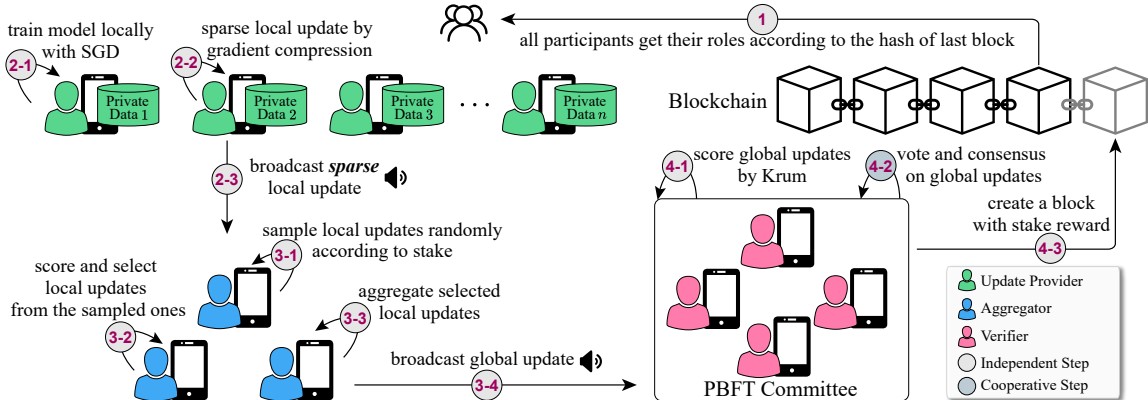

**Figure 1: The detailed processes of BlockDFL in one round of communication, from ① to ④.**

At the start of each communication round, each participant is randomly assigned with a role based on the hash of the last block as in [26] (process ①). Then, update providers train local models with a stochastic gradient descent (SGD) algorithm on their own training set and sparse the local updates by gradient compression before broadcasting them to aggregators (process ②). Each aggregator continues to receive local updates until a certain number of local updates are obtained, and then starts the aggregation independently (process ③). An aggregator first samples a certain number of local updates from the received ones according to the stake of the corresponding providers. Then, it scores the sampled local updates and selects some of them to aggregate a global update which is then broadcasted to verifiers. When verifiers receive enough global updates (e.g., the global updates from the super majority of aggregators), the verification starts (process ④). Each verifier independently scores the global updates and votes for them based on the scores, so as to select one approved global update. Finally, the approved global update with its relative information is wrapped in a new block that is then broadcasted to all participants.

Each non-empty block contains five components: 1) the hash of the previous block; 2) the approved global update and the identities of the corresponding aggregator and update providers; 3) the signed votes from verifiers; 4) the stake increment of relevant participants; and 5) the identity of the creator with its digital signature of this block. BlockDFL introduces blockchain to: 1) consistently and randomly assign roles through the hash of the last block [41]; 2) synchronize global updates through newly-added blocks; and 3) distinguish contributions through the stake [42]. Participants with honest behaviors will continue to accumulate stake, making malicious participants less and less influential to the entire FL system.

## 4 DETAILED PROCESSES OF BLOCKDFL

### 4.1 Role Selection

At the start of role selection in BlockDFL, the hash value of last block $h_{-1}$ is mapped to a hash ring where each participant is assigned a space proportional to its stake as in [26]. The participant whose portion corresponds to $h_{-1}$ is selected as the first aggregator. Then, the hash value is repeatedly re-hashed to select other aggregators. When a certain number of aggregators are selected, it turns to select verifiers in the same way. When all aggregators and verifiers are

selected, the rest participants become update providers. It ensures that participants with more stakes are more likely to be selected as important roles, i.e., aggregators and verifiers. The set of verifiers, aggregators and update providers are represented by $\mathcal{V}$, $\mathcal{A}$ and $\mathcal{U}$, respectively. The number of verifiers $|\mathcal{V}|$ and the number of aggregators $|\mathcal{A}|$ are both hyper-parameters set before BlockDFL starts. As illustrated in Section 5.3, the efficiency of BlockDFL is mainly related to the number of aggregators and verifiers. Thus, $|\mathcal{V}|$ and $|\mathcal{A}|$ are recommended to be much smaller than $|\mathcal{U}|$.

In BlockDFL, roles are reassigned at the start of each round to give each participant the opportunity to contribute its local update to an FL system and defend bribery attack.

### 4.2 Local Training

In round $t$, update provider $u_i$ performs local training based on the model parameters of the previous round $\mathbf{w}_i(t-1)$ on its private training data with the SGD algorithm as:

$$\mathbf{w} - \eta \frac{1}{b} \nabla \mathcal{L}(\mathbf{x}, \mathbf{w}) \rightarrow \mathbf{w} \tag{1}$$

where $x$ is a mini-batch with $b$ samples of the training set $\mathcal{X}$ of $u_i$, $\mathcal{L}$ is the loss function and $\eta$ is the learning rate (step 2-1 in Fig. 1). Let $\mathbf{w}_i(t)$ be the model parameters after several epochs of local training. The local update $\mathbf{d}_i$ is obtained as:

$$\mathbf{d}_i = \mathbf{w}_i(t) - \mathbf{w}_i(t-1) \tag{2}$$

To protect the representation privacy of local data and reduce the communication cost, we apply top-k sparsification to the local updates as in [7]. Let $s$ be the sparse ratio, i.e., the percentage of zero elements in the sparsed local update $\mathbf{d}_i$, the update provider only transmits the $(1-s)|\mathbf{d}_i|$ elements of $\mathbf{d}_i$ with the largest absolute value (step 2-2 in Fig. 1). To avoid the loss of accuracy, the rest elements are kept locally and accumulated to the next local training of the participant as in [21]. The sparse local update is digitally signed and broadcasted to aggregators (step 2-3 in Fig. 1).

### 4.3 Aggregation

When an aggregator has collected a certain number of local updates, the aggregation starts where each aggregator performs the same process independently. Let $\mathcal{D}$ denote the set of local updates received by an aggregator and $c$ be the number of local updates

that a global update must contain. There are two sampling steps in aggregation. The first step is to discard most of the model updates based on the corresponding stake for relieving the computation cost of later testing. In this step, the aggregator samples $3 \times c$ local updates from $\mathcal{D}$, where the probability of each local update to be selected is proportional to the stake of its update provider (step 3-1 in Fig. 1). The sampled local updates constitute a set $\mathcal{D}^S$. To make more honest participants have the opportunity to share their local updates, the stake can be log-scaled here.

The second step conducts the median-based testing to select high-quality un-poisoned local updates for aggregation. Each aggregator updates the global model in last round with local updates in $\mathcal{D}^S$ one by one and performs inference on a subset randomly sampled from its own training set (step 3-2 in Fig. 1). Then, local updates in $\mathcal{D}^S$ are ranked in descending order according to their accuracy of inference. Let $\mathcal{DM}^S$ denote the local updates before the median of sorted $\mathcal{D}^S$, the local updates for aggregation are randomly selected from $\mathcal{DM}^S$ and constitute a set $\mathcal{D}^A$ ($|\mathcal{D}^A| = c$). The probability $p_i$ of each local update $\mathbf{d}_i$ in ranked $\mathcal{DM}^S$ to be selected is:

$$p_i = \frac{\exp(q(\mathbf{d}_i))}{\sum_{\mathbf{d}_j \in \mathcal{DM}^S} \exp(q(\mathbf{d}_j))} \tag{3}$$

where $q(\mathbf{d}_i)$ is the inference accuracy of local update $\mathbf{d}_i$ on the subset of the training set held by the aggregator. Local updates in $\mathcal{D}^A$ are aggregated to a global update $G$ as:

$$G = \frac{1}{|\mathcal{D}^A|} \sum_{\mathbf{d} \in \mathcal{D}^A} \mathbf{d} \tag{4}$$

(step 3-3 in Fig. 1). The aggregated global update $G$ is then digitally signed and broadcasted to verifiers to compete for being packaged on the blockchain (step 3-4 in Fig. 1).

There are two reasons for adopting median-based testing instead of Krum to select local updates: 1) The complexity of Krum is $O(n^2)$ where $n$ is the number of model updates to be evaluated, thus, Krum may not be appropriate to score a large number of model updates such as verifying local updates [26], where $n$ is relatively large; and 2) Krum suffers from the non-IIDness of training data, since non-IID data enlarge the distance between local updates. The non-IIDness has a relatively small impact on testing, as the local models trained on non-IID datasets do not lead to the inability to distinguish between specific two categories of data as poisoned models do. The experiments in Section 5.2 also confirm this.

The stake filter makes most of the local updates to be tested come from honest participants, so that the model updates before the median are un-poisoned. Median-based testing determines whether a local update is poisoned by comparing its score with that of the others, instead of relying on a manual threshold [40] or a baseline validation model which may be hard to obtain in real world to judge whether an update is malicious. Therefore, this process is reliable and applicable.

## 4.4 Verification and Consensus

In order to uniquely elect one suitable global update in each round, we simplify PBFT [4] for decentralized FL and design a voting-based verification mechanism based on the simplified PBFT, which has the following advantages: 1) It has high efficiency since the verifiers

are a small group of randomly selected participants; 2) It can deal with malicious participants and the disconnection problem, even for the leader of PBFT; and 3) It never forks.

The first selected verifier is the leader of verifiers to initiate the verification of global updates one by one in the order it determines. There are three stages in the proposed voting mechanism that each global update needs to go through, i.e., *pre-prepare*, *prepare* and *commit*. Let $\mathcal{G}$ be the set of candidate global updates in this communication round. Assuming that $G_i \in \mathcal{G}$ is the first selected global update to be verified. In the verification of $G_i$, the leader first sends a *pre-prepare* message with the digital signature of $G_i$ to the other verifiers. When a verifier receives the *pre-prepare* message, it broadcasts a *prepare* message with the digital signature of $G_i$ to all verifiers. When a verifier receives more than $\frac{2}{3}|\mathcal{V}|$ *prepare* messages, it starts the *commit* stage. In *commit*, the verifier scores each $G_i$ by Krum [3], where a lower score indicates a higher quality. Let $f$ be the percentage of malicious participants and $\mathcal{G}_i^c \subseteq \mathcal{G}$ denote the $(1 - f)|\mathcal{G}| - 2$ global updates closest to $G_i$, Krum scores $G_i$ by calculating the distance of $G_i$ to global updates in $\mathcal{G}_i^c$, as:

$$\text{Krum}(G_i, \mathcal{G}) = \sum_{G_j \in \mathcal{G}_i^c} \left\| G_i - G_j \right\|^2 \tag{5}$$

The score of the other global updates in $\mathcal{G}$ is calculated as in step 4-1 in Fig. 1. Then each verifier sends a signed *commit* message to the leader containing the vote to $G_i$. Only the score of $G_i$ surpasses that of 2/3 global updates can $G_i$ be voted affirmatively, as:

$$\begin{cases} 1 & \text{if } \sum_{G_j \in \mathcal{G} \setminus G_i} \mathbb{I}_{\text{Krum}(G_i, \mathcal{G}) < \text{Krum}(G_j, \mathcal{G})} \geq \frac{2}{3}|\mathcal{G}| \\ 0 & \text{else} \end{cases} \tag{6}$$

where 1 and 0 means the affirmative and negative vote, respectively. $\mathbb{I}$ is 1 when the condition is met and otherwise 0.

If the leader has received more than $\frac{2}{3}|\mathcal{V}|$ *commit* messages with the affirmative vote, the verification ends and $G_i$ becomes the approved global update of this communication round (step 4-2 in Fig. 1). Then the leader builds a block containing: 1) the elements of $G_i$, 2) the identity of the aggregator and update providers of $G_i$ and 3) the identity of the verifiers who vote for support. The block is signed by the leader and broadcasted to all participants (step 4-3 in Fig. 1). The participants listed in 2) and 3) are equally awarded with stake. However, if the number of *commit* messages with the negative vote the leader received has exceeded $\frac{1}{3}|\mathcal{V}|$, the verification of $G_i$ is finished and the leader starts the verification of another global update $G_j$. Note that in the verification of the subsequent global updates, the score of them obtained during the verification of the first global update can be directly used. If all global updates in $\mathcal{G}$ are verified but no one is approved, the leader broadcasts an empty block. When a participant receives a block, it updates the local model if the block contains an approved global update. Then, the next communication round starts.

Note that the ability of the leader to behave maliciously is limited, since the leader can broadcast a global update only if the affirmative votes of more than 2/3 verifiers are acquired. If this condition is not met, it can only choose to verify the next global update or finish the current round of communications. Thus, if the leader is malicious, what it can do to harm the system is to deny the votes of other verifiers and broadcast an empty block to delay the iteration of FL.

**Table 1: Default settings of experiments.**

| Parameter name | Value |
|---|---|
| # of aggregators & verifiers | 8 & 7 |
| Initial stake & stake increment | Uniformly 10 & 5 |
| $c$ of global updates | 5 |
| # of epochs in local training | 5 |
| Sparsity $s$ in MNIST | [90%, 92.5%, 95%, 97.5%] changes every 50 rounds |
| Sparsity $s$ in CIFAR-10 | [85%, 87.5%, 90%, 92.5%, 95%] changes every 60 rounds |

We apply Krum in verification instead of aggregation due to: 1) Its result is consistent on the same updates, facilitating the consensus on the voting result. 2) Its complexity is $O(n^2)$ with $n$ updates to be scored, and $n$ is usually larger in aggregation than that in verification of BlockDFL. Intuitively, Krum is negatively affected by the sparseness of model updates since it calculates distance between them element-wisely. But we observe that many of the indexes of the $(1 - s)|\mathbf{d}_i|$ elements with the largest absolute value in different updates may overlap, enabling to spatially distinguish the sparsed normal and poisoned updates. More details are in Section A.1.

## 5 EVALUATIONS

### 5.1 Experimental Setup

We implement BlockDFL with Python 3.8 and PyTorch 1.10 to evaluate its accuracy, poisoning-tolerance, efficiency and scalability. The experiments demonstrate: 1) BlockDFL has a comparative accuracy compared with vanilla FL and can effectively resist poisoning attacks, 2) the reason that BlockDFL can resist poisoning attacks and 3) BlockDFL works efficiently and possesses good scalability.

*5.1.1 Dataset, Model and Platform.* We select two widely-used real-world datasets, i.e., MNIST [8] and CIFAR-10 [18] to evaluate BlockDFL. For MNIST, we build a convolutional neural network with 1,662,752 parameters as in [24]. For CIFAR-10, we build a CI-FARNET with 1,149,770 parameters[1]. In local training, these models are trained by SGD with learning rate at 0.01, which decays by 0.99 after each round. The other parameter settings are as listed in Table 1 unless stated otherwise. As shown in Table 1, all participants start with 10 stake, and if they are awarded with stake, the quantified value of the stake obtained is 5. The sparsity of local updates is averaged to 93.75% on MNIST and 90% on CIFAR-10.

We run 50 participants on a Windows10 platform with an AMD Ryzen 5800 3.40GHz CPU, an NVIDIA RTX 3070 GPU. The training set is randomly distributed to participants (IID setting) or sampled in Dirichlet distribution with $\alpha = 1.0$ to build practical non-IID subsets distributed to participants as in [25]. The test set is used to evaluate global model accuracy. Each participant randomly selects 20% samples from its training set to score local update (Section 4.3).

*5.1.2 Malicious Participants.* Malicious update providers poison local updates by label-flipping attack as [22, 26]. They label number

---

[1]64C3x2-MaxPool2-Drop0.1-128C3x2-AvgPool2-256C3x2-AvgPool8-Drop0.5-256-10

1 as 7 in MNIST, and cat as dog and deer as horse in CIFAR-10, then perform local training on the poisoned training set.

To better evaluate the robustness of BlockDFL, we introduce more challenges by further granting other roles the ability to facilitate poisoning attacks instead of only update provider as [26]. For example, a malicious aggregator aggregates $c$ of the received local updates with the lowest accuracy. The $c$ local updates from the update providers with the lowest stake are not directly selected because such behavior can be easily detected, thus exposing the malicious participants. A malicious verifier will vote contrarily to an honest one, aiming at a wrong consensus result.

### 5.2 Accuracy and Poisoning Tolerance

*5.2.1 Evaluation.* We evaluate the accuracy of BlockDFL by comparing it with the vanilla FL [24] (relies on a trusted centralized server) and Biscotti [26] without DP and with the power of DP set to the lowest value as in [26]. The reason for implementing Biscotti is that it bears similarities to our work, and from our investigation in Section 5.4, it is a decentralized P2P FL frameworks based on blockchain that exhibits robust resistance against poisoning attacks and has gained widespread recognition in recent years. Comparisons with more existing frameworks are presented in Section 5.4. Model updates in vanilla FL and Biscotti are transmitted without sparsification. The relevant FL settings of Biscotti are the same as those of BlockDFL. Note that the goal of BlockDFL is not to surpass vanilla FL in accuracy, but to obtain the accuracy as close as possible to vanilla FL in a fully decentralized system, while preventing the FL system from being jeopardized by malicious participants.

We iterate BlockDFL, vanilla FL and Biscotti for 200 rounds of communication on MNIST and 300 rounds on CIFAR-10 and subject both of them to poisoning attacks with the proportions of malicious participants ranged in [0%, 60%]. We run them for five times for each proportion of malicious participants. The *average test accuracy* is calculated by averaging the *inference accuracy* by the global model on the whole test set in the last 20% rounds of these runs, i.e., the last 40 rounds for MNIST and 60 rounds for CIFAR-10. Fig. 2 presents the average test accuracy with the corresponding standard deviation of these approaches. When there is no malicious participant on IID datasets, BlockDFL achieves the average accuracy of 99.29% on MNIST while vanilla FL is 99.28%, and achieves the average accuracy of 87.41% on CIFAR-10 while vanilla FL is 87.84%. On non-IID datasets, BlockDFL is slightly inferior to vanilla FL with a very small gap. Thus, BlockDFL can achieve comparable accuracy compared with vanilla FL, which is a centralized scheme.

When facing malicious participants ($\leq 40\%$), BlockDFL keeps relatively steady average test accuracy on both datasets and with both IID and non-IID settings, while vanilla FL is severely jeopardized. The average accuracy gap between BlockDFL and vanilla FL increases with more malicious participants. Moreover, BlockDFL converges much more stably than that of vanilla FL when facing malicious participants, showing very low standard deviation of the last 20% rounds. As for Biscotti, on the simple MNIST dataset, it can defend poisoning attacks with 30% participant are malicious, however, on complex CIFAR-10 dataset, this ratio is less than 20%. Besides, DP can cause a severe drop in accuracy on complex CIFAR-10 dataset, although the power of DP is set to the lowest value as

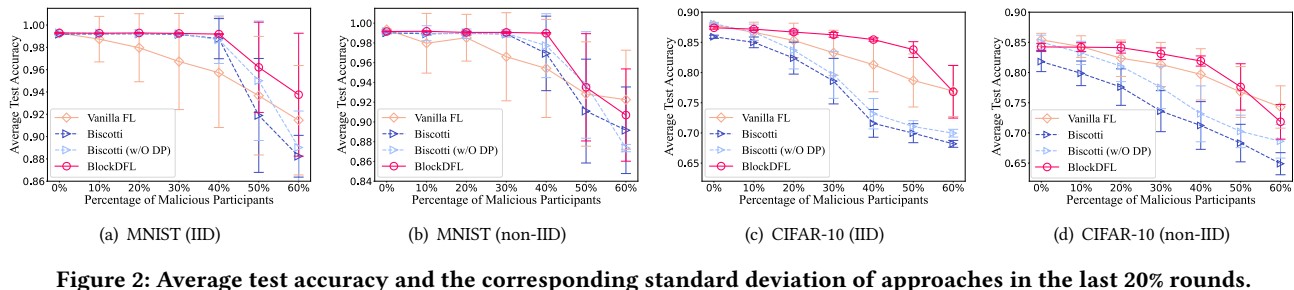

| (a) MNIST (IID) | (b) MNIST (non-IID) | (c) CIFAR-10 (IID) | (d) CIFAR-10 (non-IID) |

**Figure 2: Average test accuracy and the corresponding standard deviation of approaches in the last 20% rounds.**

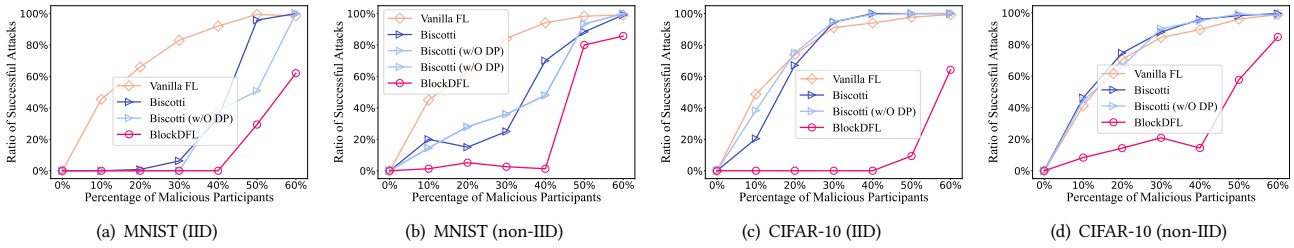

| (a) MNIST (IID) | (b) MNIST (non-IID) | (c) CIFAR-10 (IID) | (d) CIFAR-10 (non-IID) |

**Figure 3: Successful attack ratio in approaches with different percentage of malicious participants.**

in [26]. Note that on CIFAR-10, a relatively complex dataset than MNIST, the average test accuracy of BlockDFL slightly decreases with the increasing ratio of malicious participants, although it is still significantly better than that of vanilla FL. The same phenomenon also appears in [22], because when the system can defend against poisoning attacks, the training data held by malicious participants are excluded. Thus, as the ratio of malicious participants increases, the amount of data contributed to the global model decreases.

Figure 3 illustrates the ability of these frameworks to resist poisoning attacks, where successful attack ratio (SAR) means the percentage of global updates containing at least one poisoned local updates in the last 20% rounds. As illustrated, the SAR of BlockDFL is almost 0% when the ratio of malicious participants is no more than 40% on MNIST and IID CIFAR-10. On non-IID CIFAR-10, the SAR of BlockDFL is a little bit higher, but the model accuracy is not significantly jeopardized because such a small number of malicious model updates can hardly cause significant negative impact. For Biscotti and vanilla FL, it is more likely to occur that the global updates of vanilla FL contain one or more poisoned local updates. From the above, we conclude that BlockDFL is able to defend poisoning attacks when there are up to 40% malicious participants.

To present the convergence of BlockDFL, we select 6 runs in IID setting, each of which corresponds to a different dataset and different ratio of malicious participants, and plot the test accuracy after each round of communication in Fig. 4. As shown in Fig. 4(a) and 4(d), the convergence speed of BlockDFL is very close to un-poisoned vanilla FL, indicating that BlockDFL does not require more communication rounds than vanilla FL. It can be observed that in Fig. 4(b), 4(c), 4(e) and 4(f), when there exist malicious participants, BlockDFL gradually becomes immune to poisoning attacks and converges to a level close to the un-poisoned FL, while the poisoned FL diverges seriously. When there are fewer malicious participants, BlockDFL become immunes to poisoning attacks faster.

Limited by space, more discussions and illustrations on the effectiveness of BlockDFL are available in Appendix B

## 5.3 Time Consumption and Scalability

To show the efficiency and scalability of BlockDFL, we run it on MNIST with variant numbers of participants ranged in [20, 60] and record the time consumption of aggregation and verification. Since scoring local updates and scoring global updates are important steps of aggregation and verification, respectively, we also record the time consumption of the two steps. We fix the numbers of verifiers and aggregators to 4, $c$ of global updates to 3 and scale the number of participants. Since the training set on each participant is equally divided from the original dataset, the number of participants affects the number of samples in the data set for scoring local update on each participant, thus affecting the time taken for scoring local updates in aggregation. To eliminate this impact, we fix the number of the samples on each participant for scoring local updates to 150.

Fig. 5(a) presents the time spent by each process and step with a varying number of participants. We can find that the time spent by aggregation mainly lies in scoring local updates, while scoring global updates takes much less time than verification, meaning that the time of verification is mainly spent by voting cooperatively. With the changes of the number of participants, the time spent by each process keeps steady, implying a good scalability of BlockDFL.

BlockDFL achieves better efficiency than Biscotti [26], which is a relatively comprehensive framework. Biscotti is evaluated on MNIST with a model containing only 7,850 parameters, and it takes over 30 seconds for aggregation and verification when there are 40 participants as reported in [26]. While our BlockDFL is evaluated with a model containing 1,662,752 parameters on MNIST, and takes less than 3 seconds totally for aggregation and verification, which is very short compared with that of Biscotti. Attributed by 1) the fast consensus in a small group of randomly selected participants and 2) the mechanism that makes the aggregators themselves responsible for aggregation and thus removes the dependency on homomorphic commitment and secret sharing for anti-poisoning, the efficiency of BlockDFL significantly outperforms Biscotti. The latter heavily relies on homomorphic commitment and secret sharing to ensure

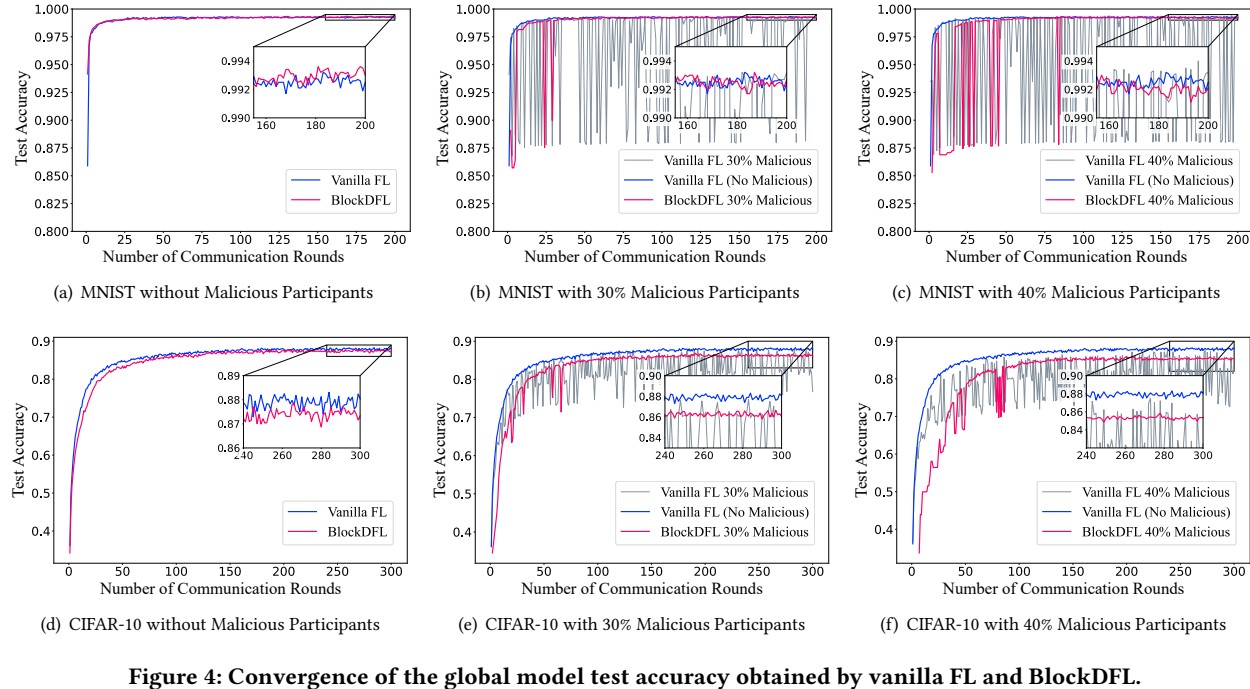

(a) MNIST without Malicious Participants

(b) MNIST with 30% Malicious Participants

(c) MNIST with 40% Malicious Participants

(d) CIFAR-10 without Malicious Participants

(e) CIFAR-10 with 30% Malicious Participants

(f) CIFAR-10 with 40% Malicious Participants

**Figure 4: Convergence of the global model test accuracy obtained by vanilla FL and BlockDFL.**

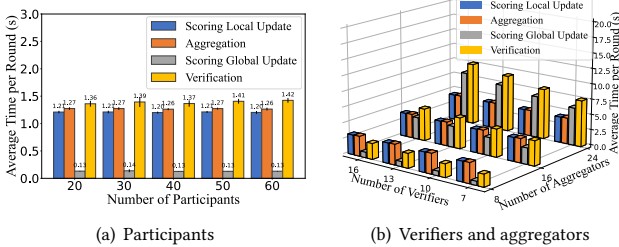

(a) Participants

(b) Verifiers and aggregators

**Figure 5: Time consumption of processes in BlockDFL with varying number of participants, aggregators and verifiers.**

the aggregation not compromised by malicious participants. Thus, BlockDFL obtains excellent efficiency and scalability.

To clarify how the numbers of aggregators and verifiers affect the efficiency of BlockDFL, we run BlockDFL with different numbers of aggregators and verifiers. As shown in Fig. 5(b), the time consumption of verification together with scoring global updates are mainly related to the number of aggregators. When the number of aggregators grows, both of them increase, since the global updates that need to be scored and the average number of votes required to select the final global update also increase. But they are less affected by the number of verifiers. As for aggregation, it almost keeps steady with different numbers of aggregators and verifiers, since the aggregators work independently.

## 5.4 Multi-dimensional Qualitative Comparisons

Since BlockDFL is designed for decentralized P2P FL, we investigate some existing blockchain-based decentralized FL frameworks without the reliance on a global trust authority or trusted servers,

i.e., FL frameworks for the **fully decentralized P2P setting**, and compare them in six dimensions in Table 2: 1) **Privacy**: The privacy protection on the basis of FL. 2) **Anti-Poisoning**: Ways to defend poisoning attacks. 3) **Poisoning Tolerance**: The percentage of malicious participants the framework can tolerate. 4) **Efficiency Optimization**: Ways to optimize efficiency. 5) **Consensus & Fork-preventing**: The consensus and whether it can prevent forking problems of blockchain. 6) **Dataset**: The datasets utilized for evaluation, together with the data distribution among participants.

As shown, different P2P frameworks introduce different mechanisms to partially solve the problems faced by FL, i.e., security, efficiency and privacy. However, they do not address these issues uniformly, e.g., address poisoning attacks but ignore privacy protection or protect the privacy but fail to prevent poisoning attacks. Particularly, they often neglect to optimize the system efficiency.

As for the ability of poisoning tolerance, existing frameworks can only defend against poisoning attacks when the proportion of malicious participants is less than or equal to 30%, while BlockDFL can effectively defend poisoning attacks when 40% of the participants are malicious. Moreover, the evaluations of existing frameworks need to be further improved, i.e., 1) most of the existing frameworks are only evaluated on a simple dataset, i.e., MNIST (Wisconsin breast cancer is even simpler, and Credit Card is also so simple that a logistic regression model can handle it [26]), while BlockDFL is evaluated on both MNIST and a relatively complex dataset, i.e., CIFAR-10; and 2) existing frameworks are only evaluated with the data among participants are IID, ignoring the nature of non-IID data in FL, while BlockDFL is evaluated on both IID and non-IID data. These enhance the persuasiveness of evaluation results for BlockDFL. From these comparisons, BlockDFL outperforms all the existing fully decentralized P2P FL frameworks in terms of poisoning tolerance.

**Table 2: Multi-dimensional Comparisons of Existing Decentralized P2P Federated Learning Frameworks based on Blockchain**

| Approach | Privacy | Anti-Poisoning | Claimed Maximal Poisoning Tolerance | Efficiency Optimization | Consensus & Fork-preventing | Dataset |
|---|---|---|---|---|---|---|
| LearningChain [5] | DP | *l*-Nearest Aggregation | 10% Malicious | ✗ | PoW ✗ | Synthetic & Wisconsin breast cancer [10] & MNIST (all IID) |
| BlockFL [17] | DP | ✗ | ✗ | ✗ | PoW ✗ | Unspecified |
| BEMA [32] | ✗ | Krum + Multiparty Multiclass Margin | 20% Malicious | ✗ | PoW ✗ | MNIST (IID) |
| DeepChain [34] | Homomorphic Encryption | ✗ Auditing | 0% Only Traceability | ✗ | Algorand ✓ | MNIST (IID) |
| Biscotti [26] | DP + Secure Aggregation | Krum | 30% Malicious | ✗ | PoF ✗ | MNIST & Credit Card [10] (all IID) |
| **BlockDFL (ours)** | Gradient Compression | Median-based Testing + Krum | **40% Malicious** | PBFT in a small group | PBFT-based Voting ✓ | MNIST & CIFAR-10 **(IID and non-IID)** |

## 6 RELATED WORK

In recent years, there are many studies about blockchain-based FL. Some of them are inapplicable in fully decentralized P2P contexts due to 1) relying on a global trust authority or trusted servers [14, 20, 37, 45], or 2) relying on a group of pre-set miners or special nodes with higher authority to run the consensus [6, 9, 16, 29] where the powers of participants are not equal. These frameworks may still face high latency due to long-distance transmission, because the pre-set nodes may not be geographically close to the participants.

There are also some studies focus on fully decentralized P2P FL based on blockchain, and some of them integrate additional protections for solving the privacy and security concerns. To provide better privacy protection, existing frameworks add DP noise with certain power on local models [26], or leverage homomorphic encryption [2, 34] and secure aggregation [23, 26] to prevent the model update of a specific participant from being intercepted by attackers. To filter out poisoned model updates, existing frameworks conduct Krum [26] or *l*-nearest aggregation [5] on received local updates before aggregation, or discard model updates with accuracy lowering than a pre-defined accuracy threshold [39], or leverage auditing to track the history of behaviors [2].

These frameworks perform well in different scenarios. However, in a fully decentralized scenario without mutual trust, existing frameworks still need to be further improved. First, some of them only solve part of the problems of FL, i.e., addressing poisoning attacks but ignoring the privacy protection [32, 34, 39] and protecting the privacy but failing to prevent poisoning attacks [2, 23]. Second, some of the technical selections bring negative impact on FL, e.g., although DP is able to protect privacy with the guarantees of mathematical proof from the perspective of data reconstruction and membership inference, etc., it imposes a significant loss of accuracy for protecting complicated models [33]. Homomorphic encryption brings too much computation overhead, making it unsuitable for models with relatively large numbers of parameters. Secure aggregation brings heavy overhead of computation and communication, which limits the efficiency and scalability of FL frameworks based on it. Finally, the resistance to poisoning attacks

of existing frameworks could be enhanced. As presented in Table 2, existing frameworks can only tolerant up to 30% malicious clients for IID data, while BlockDFL can defend against poison attacks when there are 40% malicious clients for both IID and non-IID data.

In addition to poisoning resistance, BlockDFL is also efficient since: 1) the verification of local updates is fast since the stake-based filtering mechanism drops many local updates, 2) the PBFT-based voting for global update election works efficiently since the verifier committee is composed of only a small group of participants and 3) the communication cost of transmitting model updates is further lowered by gradient compression. It is worth noting that the proposed voting mechanism for the consensus on global updates does not need to perform meaningless hash calculations like proof-of-working (PoW) consensus and never forks as in [5, 6, 17, 26, 32].

## 7 CONCLUSIONS

In this paper, we propose BlockDFL, an efficient fully decentralized P2P FL framework, which leverages blockchain to force participants to behave correctly. To efficiently reach the consensus on the appropriate global update, we propose a PBFT-based voting mechanism conducted among a small group of participants randomly selected in each round. To utilize high-quality model updates, we propose a two-layer scoring mechanism that measures local and global updates by median-based testing and Krum, respectively. The combination of the two mechanisms helps BlockDFL uniquely select a high-quality global update in each round, while preventing the FL system from being poisoned. To protect the privacy of data in terms of data representation, we introduce gradient compression, and experimentally demonstrate that gradient compression can be integrated into BlockDFL without affecting the effectiveness of Krum and the accuracy of a global model, while protecting privacy and reducing communication overhead. Experiments conducted on two widely-used real-world datasets demonstrate that BlockDFL can defend the poisoning attacks and achieve both high efficiency and scalability. Specially, when 40% of the participants are malicious, BlockDFL can defend poisoning attacks, which outperforms existing fully decentralized FL frameworks based on blockchain.

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

# A  SUPPLEMENTARY EXPERIMENTS

## A.1  Distance between Sparsed Model Updates

As introduced in Section 4.4, we observe that when the gradient compression is introduced, some of the indexes of the transmitted elements with the largest absolute value in different updates still overlap, enabling to spatially distinguish the normal model updates and the poisoned ones.

We randomly split MNIST into 50 subsets with the same number of data samples (IID settings) and poison 15 of them by label-flipping attack as introduced in Section 5.1, in order to simulate the situation that there are 30% malicious participants in a FL system. We train 50 CNN with 1,662,752 parameters introduced in Section 5.1 for 5 epochs, where each CNN is trained on one of the subsets. Then, we visualize the 50 model updates by t-SNE [30] in Fig. 6(a), where a blue dot indicates a poisoned update and a red dot indicates a normal update. We can observe that the 50 original updates form two clusters that one cluster is composed of normal updates and the other cluster is composed of poisoned ones.

We then sparse the 50 original model updates to 90% sparsity and visualize them by t-SNE in Fig. 6(b). As shown, the sparse updates can still form two clusters with clear boundaries according to whether they are poisoned. Note that there are some poisoned updates mixed with normal updates in Fig. 6(a), while that does not exist in Fig. 6(b). Because there are a huge number of parameters in each model, and some unimportant parameters may have a negative impact on distinguishing whether the model is poisoned or not. Sparsification helps to focus only on the most important parameters, which may reduce the negative impact of unimportant parameters on the distance calculation. It may help to improve the effectiveness of Krum, as discussed in Section 2.1 and experimentally demonstrated in Section 5.2. Therefore, we can conclude that Krum and gradient compression can be simultaneously integrated in a framework without affecting each other.

## A.2  Gradient Compression versus Data Reconstruction

Gradient compression believes that the contribution of elements in the gradient to the model accuracy is different. Elements with larger absolute values usually contribute more to the model accuracy. In distributed machine learning (DML), it can reduce communication overhead by only transmitting the elements with relatively large absolute values. Model updates in FL are similar to gradients in DML, so gradient compression can be applied to FL directly.

Although gradient compression is not proposed for privacy protection, it can effectively prevent training data from being reconstructed from gradients [27, 28, 48]. Taking one of the famous model inversion attacks, Deep Leakage from Gradients (DLG) [48], as an example, gradient compression destroys the optimization objectives of DLG attack by discarding many details of the model updates, making it hard for DLG attack to obtain enough information for reconstructing the training data from sparse gradients.

In [48], a series of experiments are conducted to evaluate the performance of DLG attack under different degree of gradient sparsity (ranged in [1%, 70%]), drawing a conclusion that DLG can tolerant up to 20% sparsity of gradients. When the sparsity exceeds this

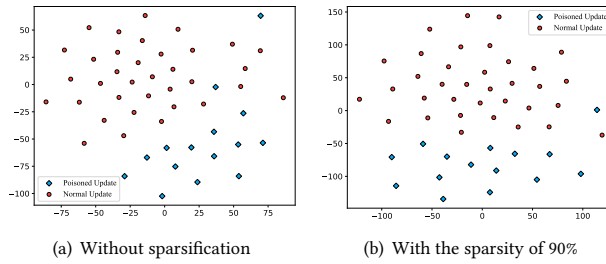

(a) Without sparsification  (b) With the sparsity of 90%

**Figure 6: 2-dimensional visualization of model updates on MNIST, where each point represents a model update.**

threshold, reconstructed images are almost not visually recognizable. We also conduct DLG attack to the model in [48] on CIFAR-10 and MNIST with different sparsity of model updates, respectively, to demonstrate this. For each image, we iterate DLG model for 300 rounds and record the result with the lowest Mean Squared Error (MSE) between the reconstructed image and the original one. Fig. 7 presents the lowest MSE and the corresponding reconstructed images with different sparsity, where the first and next row show the results on an image of CIFAR-10 and MNIST, respectively. It is observed that as the sparsity gets higher, the visibility of the reconstructed images by DLG attack gets worse. The results are consistent as reported in [48]: when the sparsity exceeds 20%, the reconstructed images are hard to be visually recognized.

However, it cannot fully guarantee that the information will not be leaked if the reconstructed images are only visually unrecognizable. To better illustrate the effectiveness of gradient compression to defense DLG attack, randomly generated images are also introduced as a reference. As shown, for CIFAR-10, when the sparsity exceeds 70%, the MSE of reconstructed image is similar to the randomly generated one, and the corresponding threshold is about 90% for MNIST. Thus, we conclude that gradient compression can effectively defense the DLG attack with the ratio of sparsity over 70% for CIFAR-10 and 90% for MNIST.

In our experiments of BlockDFL, the sparsity of model updates is more than 85% for CIFAR-10 (averaged to 90%) and more than 90% for MNIST (averaged to 93.75%) to ensure the representation privacy of local training data. Generally, on the same dataset, the more parameters a model has, the greater sparsity can be employed. Because deep learning models are usually over-parameterized, which has also been empirically validated through various studies on model quantization [11, 35].

# B  DISCUSSIONS ON THE EFFECTIVENESS OF BLOCKDFL

In this section, we provide some discussions on the reasons that BlockDFL is effective to defend against poisoning attacks.

Generally, PBFT can tolerate $f$ malicious participants when there are $(3f + 1)$ participants. However, as shown in Fig. 2, we experimentally demonstrate that BlockDFL can tolerant 40% malicious participants. Such enhancement origins from the accumulation of stake held by honest participants. As described in Section 4.4, the voting mechanism can produce a non-empty block only if the super

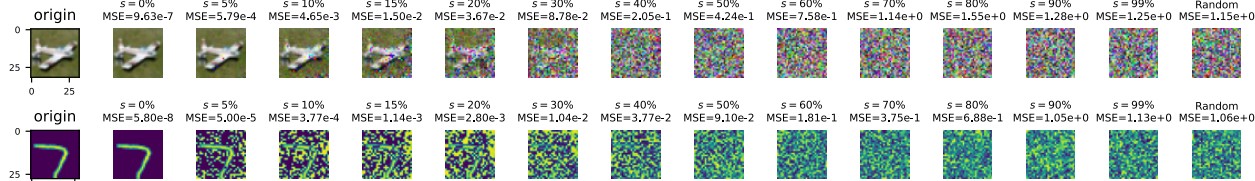

**Figure 7: MSE of images reconstructed by DLG attack [48] with different ratio of sparsity $s$ on CIFAR-10 and MNIST.**

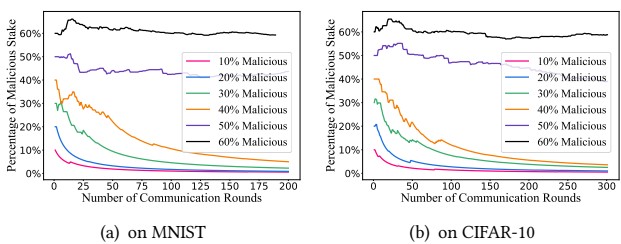

(a) on MNIST          (b) on CIFAR-10

**Figure 8: Changes in proportion of stake held by malicious participants as the rounds go on.**

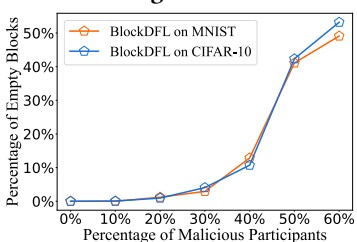

**Figure 9: Percentage of empty blocks in BlockDFL.**

majority of verifiers (over 2/3) are honest or the super majority of verifiers are malicious. When a non-empty block is created, all

the verifiers who have voted positively obtain stake. The situation that over 2/3 verifiers are honest is more likely to occur than the situation that over 2/3 verifiers are malicious when the number of honest participants is larger than that of malicious ones. Thus, the proportion of stake held by honest participants gradually increases as blocks continue to be generated, making the situation that 2/3 verifiers are honest more and more likely to occur. Fig. 8 present the proportion of stake held by malicious participants decreases as the FL rounds on IID MNIST and CIFAR-10. As shown in Fig. 8, when there are no more than 40% malicious participants, the proportion of stake held by malicious participants decreases as the rounds go on, meaning that malicious participants are increasingly unlikely to be elected as aggregators or verifiers, and thus less likely for a successful poisoning attack to occur. More specifically, the fewer malicious participants, the more stably the proportion of malicious stake declines. Note that the stake can also helps to arbitrating the benefits of participants [47]. Fig. 9 shows the percentage of empty blocks. An empty block means that neither honest nor malicious participants occupy more than 2/3 among verifiers in the corresponding round, resulting in a failed consensus of voting. The percentage of empty blocks rises with the increase of malicious participants, meaning that when there are more malicious participants, it is more difficult to reach a consensus due to the conflicts between honest and malicious participants.

