# OpenReview forum: "BlockDFL: A Blockchain-based Fully Decentralized Peer-to-Peer Federated Learning Framework"
_ACM.org/TheWebConf/2024/Conference — TheWebConf24 Oral_

### Official Review · Reviewer_qDqw · 2023-11-05

**Novelty:** 6
**Technical Quality:** 6

**Review:**

### Summary

In this paper, the authors propose and implement a blockchain-based federated learning framework, named BlockDFL. Specifically, it addressed not only the centralized issue in the traditional federated learning, but also the privacy leakage and poisoning attack issues in the untrust decentralized model. The experimental results prove that, compared to the traditional federated learning, BlockDFL can also achieve good efficiency and scalability. Moreover, compared to other decentralized solutions for federated learning, BlockDFL can resist a higher proportion of malicious users while achieving enough efficiency on non-naive datasets.

### Strength

- This paper proposes BlockDFL, a decentralized framework for federated learning. It considers the inherent issues, i.e., malicious nodes, privacy leakage, and the efficiency issue, and proposes the corresponding solutions for these issues.
- The experiment is comprehensive, quantitatively evaluating the tolerance of the malicious nodes and the efficiency and the scalability of BlockDFL. Moreover, the authors also qualitatively compare the BlockDFL with the other decentralized federated learning solutions.
- Writing quality is good, and the paper is quite readable.

### Weakness

- Some key parts require more explicit discussion. For example, there need a section to discuss the `threat model` to explicitly illustrate the abilities of malicious users in federated learning, and a `threat to validity` to discuss the limitation of this work.
- The experiment can be further improved. For example, for the proposed three issues, the protection against privacy leakage is only discussed in text, instead of by a quantitative experiment like the other two.

### Comments

Except for the main concerns I raised above, here are some minor ones.

At L286, the authors say “The participant whose portion corresponds to $h_{-1}$ is selected as the first aggregator”. It is not clear that what is `corresponds to`. This should be clarified.

In Section 5.2 and 5.3, the authors focus more on the description of the experimental results. However, I think the authors should pay some attention on analyzing the results, like why such a distinction exists between BlockDFL and other solutions. This can make the conclusion more convincing.

In Fig. 3, 4, and 5, the authors should offer different legends and line types for different candidates to make sure readers can distinguish them even in monochromatic (black and white) print.

Some typos and formatting issues exist:

- L177: “Such kind of risk” -> “Such a kind of risk”;
- L424: “denote the” -> “denotes the”;
- L537: remove the subsubsection title.

**Questions:**

Please refer to the `Review` part.

**Ethics Review Description:**

-

**Reviewer Confidence:**

3: The reviewer is confident but not certain that the evaluation is correct

**Scope:**

4: The work is relevant to the Web and to the track, and is of broad interest to the community

---

### Official Review · Reviewer_eALM · 2023-11-06

**Novelty:** 5
**Technical Quality:** 4

**Review:**

### Summary:
The paper proposes BlockDFL, an approach to address issues prevalent in other federated learning (FL) approaches, such as the risk of data poisoning and data reconstruction attacks. It relies on a PBFT-based voting mechanism as well as a two-layered scoring concept. Further, it periodically assigns participants into one of three roles based on the hash of the most recent block. The paper compares BlockDFL to traditional (not fully decentralized) FL in terms of accuracy and achieves comparable results. The paper further stresses the superiority of BlockDFL over other approaches when a large number of malicious users are involved (BlockDFL can tolerate up to 40% of malicious users and still provides reasonable accuracy).

### Pros:
+1: Well-written paper with easy-to-understand illustrations and figures

+2: Extensive evaluation, which includes comparisons to related work

### Cons:
-1: Non-blockchain-based decentralized FL approaches are not really considered

-2: Details on the deployment and associated overhead/costs is not discussed

The paper is well-written, and I was able to follow the presentation despite not being an expert in federated learning. The presentation, illustration, and comparison of related work create a nice reading experience, which motivates to read on. Thanks for doing a great job in this regard. Additionally, I like that the performance of the approach is also compared to a "traditional", i.e., not fully decentralized, FL approach.

### Detailed Comments:

#### -1: Other Related Work
Apart from comparing the accuracy to a not fully decentralized FL approach, the paper only considers blockchain-based FL approaches when discussing related work. In my opinion, this focus weakens the papers since non-blockchain-based approaches, which do not make use of a centralized aggregator/trusted party, have been proposed in the past. As a result, the paper is currently not presenting the full picture. This limitation also applies to the nicely curated comparison of related work (Table 2). To name a few, the authors could take a look at DOIs 10.1109/TSIPN.2022.3151242 and 10.1038/s41467-023-38569-4, but I am confident that there are more relevant approaches out there. Hence, moving forward, I would expect the authors to also consider this angle.

#### -2: Deployment Discussion
The design of BlockDFL builds on blockchain technology for its operation. However, from my point of view, the paper omits certain aspects related to this design choice, making it challenging to holistically assess the paper. For example, the paper does not state what the properties of the underlying blockchain are, e.g., permissioned or permissionless. Moreover, it fails to explicitly state who operates the blockchain. I believe that the entities participating in the FL are also the operators of the blockchain. However, this information is never presented, as far as I noticed.
On a related note, the evaluation section never looks into the overhead that is being introduced by building the design on a blockchain. While it provides some numbers on the performance in Figure 5, an analysis of the storage overhead is missing. I would like to know how quickly the storage grows over time and how realistic such numbers are/would be for real-world applications. Without these details, the system's impact cannot be fully judged.

#### Other:
- Introduction: I would like to see references to the statements in Lines 69 and 103.
- Figure 1: The authors could consider updating the figure to better stress that Steps 2-3 and 3-4 occur for each node with the selected role. At the moment, the visualization could also imply that a single update is being sent.
- Related Work: The content (and arguments) presented in this section largely overlaps with the introduction. Thus, it is somewhat repetitive. I believe that this space could better be used to present additional information (see above).
- Krum vs. Multi-Krum: The paper mixes the references of [3] and [26] throughout the paper and refers to both approaches as Krum. In my opinion, this approach is confusing. Therefore, I suggest the authors to use multi-Krum when talking about Biscotti.
- The relevance to the web is only briefly outlined at the beginning of the introduction. The authors could better stress this link throughout the paper but, at least as part of the conclusion, to put the contributions into perspective.

### Post-Rebuttal

I kindly thank the authors for responding to the reviews, providing a lot of additional information, and outlining their proposed changes.
They cleared up several aspects that have previously been missing from the manuscript to truly comprehend it.
Hopefully, the authors will be able to convincingly incorporate the proposed changes when revising their paper.

**Questions:**

What is the reason for only focusing on blockchain-based FL approaches? Is there a convincing argument to exclude other decentralized approaches from the comparison?

What is the overhead of utilizing BlockDFL storage-wise? How does it compare to related work?
Who is a participant/node of the underlying blockchain, only entities that also participate in the federated learning?
What kind of underlying blockchain is being used? What are its properties (e.g., permissioned vs. permissionless)?

**Reviewer Confidence:**

2: The reviewer is willing to defend the evaluation, but it is likely that the reviewer did not understand parts of the paper

**Scope:**

3: The work is somewhat relevant to the Web and to the track, and is of narrow interest to a sub-community

---

### Official Review · Reviewer_Md1b · 2023-11-17

**Novelty:** 3
**Technical Quality:** 6

**Review:**

**Paper summary**

The paper employs blockchain to address the untrustworthiness of federated learning (FL) participants, including both aggregator and clients. It introduces BlockDFL, which is a decentralized peer-to-peer (P2P) framework that further decentralize the current aggregator-based FL methods. It includes existing scoring and defence systems like Krum and gradient comparison to mitigate issues such as poisoning attacks and model inversion attacks.

**Strengths**

+ A framework that leverages the transparency of blockchain to address the trust in FL methods.

+ The paper is well-structured and well-written

**Weaknesses**

- Lack of technical contribution. Leveraging blockchain for decentralized FL seems an intuitive idea. The defence against poisoning and model inversion attacks is also an straight application of existing mechanisms.

- Experimental settings are questionable (see detailed comments below)

**Detailed comments**

This paper presents an intriguing approach of leveraging blockchain for decentralized FL. The soundness of the approach is reasonable, given that all employed mechanisms are mature. The paper is well-written. Below I elaborate on the weaknesses listed above.

***Significance***

The paper seems falling short in demonstrating a significant technical contribution. Leveraging blockchain for decentralized FL appears intuitive, and the defense against poisoning and model inversion attacks seems to be a straightforward application of existing mechanisms. A more profound exploration of novel contributions, algorithms, or methodologies would enhance the paper's scientific impact.

***Evaluation***

The choice of small-scale datasets, i.e., MNIST and CIFAR-10, raises concerns about the generalization of BlockDFL to complex real-world scenarios.

No FHE- or MPC-based solutions are selected as the baseline (Section 5.3). Given that these solutions are considered among the most costly, it is imperative to benchmark BlockDFL against them. A comprehensive evaluation against them would provide a clearer understanding of the relative trade-offs in terms of computational overhead and security.

**Questions:**

See my comments above.

**Reviewer Confidence:**

4: The reviewer is certain that the evaluation is correct and very familiar with the relevant literature

**Scope:**

3: The work is somewhat relevant to the Web and to the track, and is of narrow interest to a sub-community

---

### Official Review · Reviewer_2pGz · 2023-11-20

**Novelty:** 3
**Technical Quality:** 4

**Review:**

This paper proposes a blockchain-based fully decentralized P2P framework for FL, called BlockDFL. It takes blockchain as the foundation, leveraging the proposed PBFT-based voting mechanism and two-layer scoring mechanism to coordinate FL among peer participants without mutual trust.

pros:
1. This paper evaluates the prposed framework on two real-world datasets.
2. This paper has a good structure.

cons:
1. How to theoretically prove BlockDFL could prevent the global model from being jeopardized by poisoning attacks and prevent the private training data from being revealed?

2. It is not reasonable to claim that BlockDFL can resist poisoning attacks when there are up to 40% malicious participants for both IID and non-IID data, as the result is not theoretically proven.

3. Two works used for comparison lack representativeness. For example, why not compare the proposed solution with the original FL?

4. In fig.2 (b) and (d), when facing 60% of malicious participants, the average accuracy of vanilla FL is higher than the proposed framework. It is better to explain this result.

5. When evaluating timing consumption and scalability, why do the authors only use one dataset? Meanwhile, the authors do not compare the proposed framework with vanilla FL.

**Questions:**

see cons.

**Reviewer Confidence:**

3: The reviewer is confident but not certain that the evaluation is correct

**Scope:**

3: The work is somewhat relevant to the Web and to the track, and is of narrow interest to a sub-community

---

### Official Review · Reviewer_KebL · 2023-11-22

**Novelty:** 4
**Technical Quality:** 5

**Review:**

This paper proposes BlockDFL as a fully decentralized, i.e. peer-to-peer-based, framework for federated learning that has a claimed resilience against poisoning attacks when up to 40% of the participants providing updates to the federated machine-learning model are malicious.
The authors make use of a blockchain that is updated via PBFT by a committee that is newly selected in each communication round.
BlockDFL uses the blockchain to draw randomness for reassigning the roles of update providers, aggregators, and the verifiers constituting the PBFT committee, to manage participants' stakes for contributing honestly to the learned model, and to disseminate global model updates.

The authors provide an overall convincing approach for a P2P-based federated-learning architecture and sensible evaluate their proposed design, yielding competitive results compared to related approaches.
However, there are some points worthwhile addressing:

- ~~Relevance of the paper to TheWebConf's Systems track seems like an afterthought that is briefly motivated in the introduction but never picked up again.~~ (The authors have addressed this concern during the rebuttal and I expect that they will update their manuscript in their interest of pinpointing the relevance of their contributions). The paper generally reads as if it was addressed to a different audience since preliminary knowledge of concepts from federated learning is mandatory and Section 2 is kept at a minimum. For instance, Krum and its application plays a central role in distinguishing BlockDFL from other approaches; hence, the authors should provide some technical background to aid the reader going forward.
    - In this regard, the authors should consider condensing the introduction to what is necessary to motivate the approach, and move technical discussions to Section 2 as much as possible.
- Section 2 (Preliminaries) partly antedates design choices of BlockDFL, which should be restricted to Sections 3 and 4, respectively.
- The paper should better reflect the authors' actual contributions, especially in contrast to the seemingly very similar approach of Biscotti [26]. To this end, I suggest:
    - Move Section 6 (Related Work) to the front, before the current Section 3, so that the following sections can better communicate the changes made over [26].
    - Currently, Sections 5.4 and Section 6 have a very similar theme (albeit with different scope) anyway. Maybe the authors could use the current Section 6, when moved, to better motivate the current research gap and the design goals later on.

**Minor comments:**

- Section 1: "It can also record stake for monetary reward ..."; the relation between both parts of that sentence is unclear.
- Section 1: "For example, protecting the privacy ..." is not a full sentence.
- Section 2.1: "For example, Krum [3] regards the model model updates [that?] significantly differ..."; the word "that" seems to be missing here.
- Section 5.2: That the portion of malicious participants is increased by 10% at a time (between 0% and 60%) is only implicit when taking Figures 2 and 3 into account.

Update: I acknowledge that I have read the authors' rebuttal comments.

**Questions:**

- This is likely relevant to all proof-of-stake-based approaches, but what effectively happens if the verifiers disagree and do not reach consensus of the round's block? What happens to the provided stakes then and how are the stakes locked in the first place?
- Regarding Section 5.1: How would non-uniform stakes, i.e. different weights for potentially malicious participants, affect the accuracy of BlockDFL, and how sustainable are attacks based on the attackers losing stake in each round?
- Section 5.2: What does the "lowest power of DP" mean? Weak protection (large $\varepsilon$) or small $\varepsilon$ (better protection)?
- Section 5.2: Is there any intuition why BlockDFL achieves a tiny better accuracy than centralized FL (99.29% vs. 99.28%) even when attackers are absent?
- Section 5.2: Is there an explanation for the dips of successful attacks for a 40%-attacker in the non-IID datasets (Figures 3b and 3d)?

**Ethics Review Description:**

-

**Reviewer Confidence:**

2: The reviewer is willing to defend the evaluation, but it is likely that the reviewer did not understand parts of the paper

**Scope:**

2: The connection to the Web is incidental, e.g., use of Web data or API

---

### Decision · Program_Chairs · 2024-01-22

**Decision:**

Accept (Oral)

**Comment:**

The paper received 5 reviews. Generally, the reviews were positive, with some minor concerns. The authors engaged with the reviewers effectively to address their concerns during the rebuttal. After the discussion phase, the final recommendations were 2 accept, 2 weak accept and 1 borderline. After reviewing the reviews and discussions, I am recommending acceptance. The paper makes some solid contributions and the issues raised can be addressed during the camera-ready phase.